# The Relationship between Lifestyle Behaviors and Mental Illness in Women in College

**DOI:** 10.3390/nu16142211

**Published:** 2024-07-10

**Authors:** Noelle Armstrong, Ziyang Fu, Kathleen Woolf

**Affiliations:** 1Department of Nutrition and Food Studies, Steinhart School of Culture, Education, and Human Development, New York University, New York, NY 10003, USA; na1889@nyu.edu; 2Department of Applied Statistics, Social Science, and Humanities, Steinhart School of Culture, Education, and Human Development, New York University, New York, NY 10003, USA

**Keywords:** mental illness, college age, diet quality, sleep, physical activity, lifestyle behaviors

## Abstract

Women, particularly those in college, have the highest prevalence of any mental illness (MI), which negatively impacts social connection, academic performance, and health. Research into alternative treatment methods suggests that lifestyle behaviors are safer and more effective than medication at reducing MI symptoms. This study explores the relationship between lifestyle behaviors and MI in college women through an online survey. The survey included a food frequency questionnaire (Diet History Questionnaire II) and questions about lifestyle behaviors, symptoms of MI, and sociodemographic information. Diet quality was calculated using the Healthy Eating Index (HEI), and MI was assessed using the Depression, Anxiety, and Stress Scale-21. Independent sample *t*-tests, ANOVA, and chi-square analyses were conducted. A total of 222 women completed the survey. Overall, diet quality was poor, with a mean HEI of 62.6 ± 10.4. No significant differences or associations were found between MI groups for total HEI score, sleep, or tobacco use. However, alcohol use was significantly associated with MI (*p* = 0.049). Individuals with fewer concurrent unhealthy lifestyle behaviors had less severe symptoms of depression (*p* = 0.009) and anxiety (*p* < 0.001) compared to those with more unhealthy lifestyle behaviors. In this study, alcohol use was the only independent lifestyle behavior associated with symptoms of MI. We also found an association between the number of concurrent unhealthy lifestyle behaviors and depression and anxiety symptoms. Future research should explore the benefits of a healthy lifestyle on MI in a more diverse sample of women.

## 1. Introduction

Depression and anxiety are among the top 10 contributors to overall burden of disease and disability worldwide [1]. Young adults, aged 18–25, comprise the group with the highest prevalence of any mental illness in the United States [1,2]. Approximately 12.6 million young adults reported having any mental illness in 2022, over 1 million more individuals compared to 2021 [2,3]. Additionally, women, among all age groups, have a higher prevalence of mental illness compared to men [1]. Furthermore, individuals who are enrolled in college or university experience mental illness at an even higher rate compared to those who are not [4]. Mental illness experienced in college can not only negatively impact social connection, academic performance, and health status but can also persist throughout adulthood and lead to harmful long-term impacts on morbidity and mortality [4,5,6].

Stress, anxiety, and depression, a few of the most common mental health issues among college students, have been attributed to a variety of factors, including the social, academic, and economic pressures of emerging adulthood [7,8]. College and university students experience a substantial change in their physical and social environments and deal with high-stress situations with newfound autonomy [9]. Related to this novel independence, students also experience a shift in lifestyle behaviors, which can include an individual’s diet, sleep habits, alcohol consumption, tobacco or nicotine use, and physical activity, among others [10].

College students start to develop individual dietary habits and eating behaviors separate from their parents and begin to prepare meals for themselves [11]. Research suggests that students who live away from their families consume less fruits, vegetables, legumes, and fish than those who live with their families [12]. Diet quality typically suffers during this life stage with marked increases in access to energy-dense foods, irregular mealtimes, and skipped meals [11,12]. College students tend to rely less on fresh foods and more on pre-prepared meals, take-out, and fast foods [13]. Young adults attending college or university are often ill equipped to deal with the shift in mealtime responsibility. Students tend to lack culinary skills, basic nutrition knowledge, and food preparation experience, and most college campuses do not provide these resources [14]. These phenomena are especially true for young women, who also experience disordered eating and body dissatisfaction at higher rates than men of the same age [15,16,17]. 

Sleep also suffers during this life stage [18]. Poor sleep quality is a common complaint among college students, as they often neglect sleep when dealing with the pressures of college life [18]. College students may also have difficulty developing coping mechanisms, contributing to poor sleeping habits [18]. Interestingly, research has also highlighted that women in college are more likely to experience sleeping problems compared to men [19]. 

Alcohol and tobacco or nicotine use among college students are other public health concerns. College represents a time where alcohol intake is at a lifetime high, with research suggesting that over 30% of women in college are drinking past the binge threshold [20,21,22]. Similarly, tobacco or nicotine use typically begins between the ages of 18 and 25, with over one third of students reporting using tobacco or nicotine products for the first-time in college [22,23,24,25]. Introduction of these behaviors during college increases the risk of use and misuse throughout adulthood [22,23,24,25].

Diet quality, sleep, alcohol, and tobacco or nicotine use have been independently associated with mental health in both young women and college students [18,21,26,27,28,29,30]. A recent meta-analysis found that consuming a high-quality diet, rich in fruits, vegetables, whole grains, and lean protein, was associated with a reduced risk of developing symptoms of depression [27]. Furthermore, a 2022 systematic review found similar results: students who consumed more fruits, vegetables, nuts, and fish reported fewer symptoms of mental illness, while students who consumed more refined grains, processed meats, desserts, and sweetened beverages reported more symptoms of mental illness [13]. Additionally, the connection between sleep and mental illness has been studied vastly, with research supporting a bidirectional relationship between sleep and depression, with poor sleep quality associated with symptoms of depression and depression associated with poor sleep [18]. Both insomnia and hypersomnia have been associated with depression and are included in The Diagnostic and Statistical Manual of Mental Disorders (DSM-5-TR) as symptoms of major depressive disorder [31,32,33]. Research has also shown that mental illness in young adults is associated with an increased consumption of alcohol and more frequent heavy drinking episodes [21]. Regarding tobacco or nicotine use, multiple studies suggest a bidirectional relationship with symptoms of mental illness, specifically depression [34,35,36]. 

As much of the current research is focused on the association between mental illness and individual lifestyle components, a combined effect should be examined. For instance, studies in older adults have found that individuals with a greater number of unhealthy lifestyle behaviors were more likely to have more severe depression symptoms [37,38]. However, whether or not these associations hold true for young adults, and more specifically women in college, remains unclear. As such, more research is needed to determine the extent to which lifestyle behaviors are associated with mental illness for women in college.

Women in college represent an important population to investigate in the context of lifestyle behaviors and mental illness as lifestyle habits formed during this time typically persist throughout adulthood [9,11]. Given this pivotal time frame and the prevalence of mental illness in this population, modifiable risk factors should be identified as potential areas to target for adjunctive methods of treatment and prevention strategies. Thus, the purpose of this study was to explore the relationship between lifestyle factors (diet quality, sleep duration, alcohol consumption, tobacco/nicotine use) and mental illness (anxiety and depression) in women in college. We hypothesized that lifestyle behaviors, both independently and combined, were associated with symptoms of mental illness in college women. The purpose was accomplished with the following aims: 1. determine if individual lifestyle behaviors were independently associated with symptoms of mental illness in college women and 2. examine the combined effect of lifestyle behaviors on symptoms of mental illness in college women.

## 2. Materials and Methods

This study was approved by the New York University Committee on Activities Involving Human Subjects. The online Qualtrics survey was live between 1 April 2022 and 15 December 2023.

### 2.1. Participants

Participants were eligible to join the study if they identified as a woman, were enrolled in college or university in the United States, and reported speaking Standard American English. This study was exploratory in nature with no restrictions based on age or level of education. Eligibility was determined through two questions, “Are you currently enrolled in a college or university?” and “Do you, regardless of your sex assigned at birth, identify as a woman?”. Participants were recruited via email (college listserv), flyers on a university campus, and social media (Facebook, Reddit, and Instagram). Participation was voluntary and consent was obtained electronically prior to starting the survey. Participants were entered into a lottery to receive an Amazon gift card as an incentive for completing the survey and were additionally offered an individualized nutrient analysis based on their survey results. Participants completed an online Qualtrics survey, which included two validated questionnaires, the Diet History Questionnaire II (DHQ II), and the Depression, Anxiety, and Stress Scale-21 (DASS-21) [39,40]. Additionally, descriptive questions inquired about lifestyle behaviors, housing, employment, and school status, while demographic questions collected information including age, gender, marital status, height, weight, race, and ethnicity.

### 2.2. Measures

The DHQ II is a validated food frequency questionnaire (FFQ) used to estimate food and nutrient intake [41,42,43]. The DHQ II includes a series of 134 food items and 8 questions about dietary supplements [39]. The version of the instrument used in this study asked participants to self-report their usual intake over the past year, including questions regarding portion size [39]. DHQ II data were analyzed using Diet*Calc software (Version 1.5.0) to determine food group and nutrient estimates. 

Diet quality was assessed using the Healthy Eating Index (HEI) [44]. The National Cancer Institute (NCI) provided SAS macros that were used to calculate HEI-2020 scores from the Diet*Calc software (Version 1.5.0) output [39]. The HEI assesses how closely the diet aligns with the Dietary Guidelines for Americans [45]. The HEI is calculated based on 13 dietary components, including 9 adequacy components, or foods needed in an adequate amount of for optimal health, and 4 moderation components, or foods that should be consumed in moderation for optimal health [44]. Participants were given a score from 0 to 100, with higher scores indicating a diet that more closely aligns with dietary recommendations and, therefore, a higher diet quality [44]. Participants were categorized into diet quality cutoffs based on the total score: 50 and below reflecting a poor diet quality, 51–80 reflecting moderate diet quality, and above 80 points reflecting good diet quality [46].

Symptoms of mental illness were assessed using the DASS-21, a validated tool (α = 0.93 [95% CI = 0.93–0.94]) consisting of three subscales measuring symptoms of depression, anxiety, and stress [40,47]. Each subscale contains 7 items, 21 items overall, which are completed using 4-point Likert scales. Scores for depression, anxiety, and stress subscales are determined by summing the relevant items and are then categorized into conventional severity labels as follows:Severity of depression symptoms: 0–9 “Normal”, 10–13 “Mild”, 14–20 “Moderate”, 21–27 “Severe”, 28+ “Extremely Severe”.Severity of anxiety symptoms: 0–7 “Normal”, 8–9 “Mild”, 10–14 “Moderate”, 15–19 “Severe”, 20+ “Extremely severe”.Severity of stress symptoms: 0–14 “Normal”, 15–18 “Mild”, 19–25 “Moderate”, 26–33 “Severe”, 34+ “Extremely severe”.

For the purpose of this paper, anxiety and depression scores were used to determine the presence of mental illness. Participants were categorized based on the presence of mental illness symptoms according to DASS-21 score, with higher scores indicating more symptom severity. Individuals were considered without symptoms of mental illness if they fell within the “Normal” classification for both anxiety and depression (scored ≤ 7 on the DASS-21 anxiety subscale and ≤9 on the DASS-21 depression subscale). Given that the DASS-21 was the grouping variable for our analysis, participants were required to complete this assessment prior to advancing to the remainder of the survey.

Lifestyle behaviors, including sleep duration and tobacco/nicotine use, were assessed using questions from the Behavioral Risk Factor Surveillance System (BRFSS). Alcohol consumption was determined using total grams of alcohol calculated from the DHQ II. Alcohol intake was categorized based on the daily recommended intake for women of ≤1 drink per day from the Dietary Guidelines for Americans [45]. Sleep duration was assessed using the question, “On average, how many hours of sleep do you get in a 24 h period?” [48]. Sleep duration was then categorized based on the National Sleep Foundation’s recommendations of 7–9 h per night for young adults [49]. Individuals who reported sleeping between 7 and 9 h were considered to have met the sleep requirements, while individuals who reported sleeping <7 h or >9 h were considered to have not met the sleep requirements. Tobacco/nicotine use was assessed using the question, “Do you currently use other tobacco or nicotine products? (e.g., electronic cigarettes, chewing tobacco)”.

Computer-generated and repeated responses were identified using reCAPTCHA and various questions embedded in the survey to detect fraud. For example, age was assessed through multiple questions, including “What is your current age (in years)?” and again, “To screen out random generated responses, what is your current age (in years)?”, and “To screen out random computer-generated responses, what year were you born?”. If ages did not match, these responses were excluded from analysis. A skip question was also utilized to screen out computer-generated responses, which included, “Please skip this question and move to the next section. Do not click on any of the choices that are labeled below. This question is used to filter random computer-generated answers.” If an answer was recorded for this question, the response was excluded from the analysis. Responses were also excluded if the reported heights and weights were not realistic (e.g., less than 4 ft tall or greater than 7 ft), if there was suspected bot activity (multiple responses starting and stopping at the exact same time), and if estimated daily energy intake was unrealistic (e.g., less than 500 kcal or greater than 5000 kcal).

### 2.3. Analysis

Statistical analyses were performed using R studio (Version 2023.12.1) [50]. Descriptive statistics were used to summarize participant characteristics with continuous variables represented as means and standard deviations and categorical variables represented as totals and percentages. *t*-tests were used to examine differences between individuals with or without symptoms of mental illness and continuous variables. Chi-square tests were used to examine associations between individuals with or without symptoms of mental illness and categorical variables. Residuals were checked when statistical associations were found. ANOVA was used to examine the differences between groups according to the number of concurrent unhealthy lifestyle behaviors. Pairwise comparisons were conducted using the Bonferroni correction post hoc *t*-test. Unhealthy lifestyle behaviors included an HEI score of less than 80, sleeping less than 7 h or more than 9 h per night, consuming more than 1 alcoholic drink per day, and current use of tobacco or nicotine products. Individuals were categorized based on the number of concurrent unhealthy lifestyle behaviors. Alpha was set at 0.05 for statistical significance. 

## 3. Results

A total of 2523 survey responses were recorded. In total, 718 responses were excluded for not meeting the eligibility criteria. A total of 763 responses were excluded due to incomplete diet data. A total of 820 responses were removed as suspected computer-generated responses: 321 responses had inconsistent ages, 294 responses started/stopped at the exact same time, 172 responses had unrealistic heights and/or weights, 24 responses had unrealistic daily calorie intake, and 9 responses were removed for answering the skip question. Analyses included participants who had complete mental health and dietary data (*n* = 222). 

### 3.1. Participant Characteristics

Full participant characteristics can be found in Table 1, which presents sociodemographic characteristics according to the presence of symptoms of mental illness (anxiety and/or depression symptoms). The findings showed that almost half (45.5%) of the sample were classified as having symptoms of mental illness, indicating that these participants were experiencing “Mild” to “Extremely Severe” symptoms of anxiety and/or depression.

In this sample of 222 women in college, the majority of participants were full-time (87.8%), undergraduate (74.3%) students. The women in this sample were mostly single (91%) and employed (64%). The mean age was 22.8 ± 4.7 years old and mean BMI was 24.3 ± 5.9 kg/m^2^, indicating an average normal body weight among the participants. The sample was predominantly White (70.3%), with approximately 15% of the sample identifying as Hispanic or Latino. A higher proportion of individuals than expected identified as Hispanic or Latino in the group with symptoms of mental illness (21.8%); a lower proportion of individuals than expected identified as Hispanic or Latino in the group without symptoms of mental illness (10.7%) (*p* = 0.026). In terms of housing status, most participants reported living with family (36.5%), alone or with roommates (31.1%), or in college housing (27.5%). No women in this sample reported homelessness. Almost one third (27.5%) of the total sample reported utilizing psychotropic medications, including antidepressant and anti-anxiety medications. A higher proportion of individuals than expected reported using psychotropic medications in the group with symptoms of mental illness (35.6%); a lower proportion of individuals than expected reported using psychotropic medications in the group without symptoms of mental illness (20.7%) (*p* = 0.012).

### 3.2. Mental Illness

Table 2 presents the categorization of depression, anxiety, and stress symptoms based on DASS-21 score. In this sample, 81 women experienced at least mild symptoms of depression, while 80 participants experienced at least mild symptoms of anxiety. Only 25 women reported experiencing mild or moderate stress symptoms.

### 3.3. Lifestyle Behaviors and Mental Illness

Table 3 summarizes the diet and lifestyle behaviors of participants according to the presence of symptoms of mental illness. According to the HEI categories, the majority of participants had moderate diet quality (84.7%), followed by poor diet quality (13.0%), and good diet quality (2.3%). Thus, improvement in dietary intake is needed by over 97% of the sample. There was no association between HEI category and symptoms of mental illness. The mean energy intake of this sample was 1896 ± 898 kcal and 30 ± 15 kcal/kg body weight. Average macronutrient intake as a percent of total calories was 49 ± 9%, 16 ± 4%, and 36 ± 3% for carbohydrate, protein, and fat, respectively. The Dietary Reference Intakes Acceptable Macronutrient Distribution Range (AMDR) for fat is 20–35% of total calories; therefore, the mean reported dietary fat intake in this sample of women exceeds the recommendations [51]. The energy and macronutrient intake of the participants did not show a significant difference according to the presence of symptoms of mental illness (*p* > 0.05).

Most participants in this sample were not tobacco or nicotine users (90.1%), had adequate sleep duration (68.9%), and consumed less than one drink per day (88.7%). The rates of tobacco or nicotine use and sleep duration were not associated with the presence of symptoms of mental illness (*p* > 0.05). However, alcohol use was associated with the presence of symptoms of mental illness. We found a higher proportion of individuals than expected reported consuming greater than one drink per day in the group with symptoms of mental illness (15.8%), while a lower proportion of individuals than expected reported consuming greater than one drink per day in the group without symptoms of mental illness (7.4%) (*p* = 0.049). 

### 3.4. Healthy Eating Index and Mental Illness

Table 4 summarizes the HEI results of the participants according to the presence of symptoms of mental illness. The average HEI score for this sample of 222 women in college was 62.6 ± 10.4, indicating that this sample had only a moderate diet quality overall. There were no statistically significant differences in the HEI total score or component scores according to the presence of symptoms of mental illness. However, no women in this sample met the recommendations for total diet quality. For individual components, whole grain and sodium intake stood out, with less than 5% of the sample meeting the dietary recommendations. Overall, the majority of women in this sample met the recommendations for whole fruits (70.3%), dark green vegetables or legumes (59.0%), total protein (58.6%), and seafood/plant protein (59.0%), while very few women met the recommendations for refined grains (20.3%), added sugars (24.3%), and saturated fats (12.2%).

### 3.5. Concurrent Unhealthy Lifestyle Behaviors and Mental Illness

Table 5 presents DASS-21 depression and anxiety scores according to the number of concurrent unhealthy lifestyle behaviors. Overall, there was a statistically significant difference in DASS-21 depression scores (*p* = 0.009) and DASS-21 anxiety scores (*p* < 0.001) and the unhealthy behavior group. We found that individuals with more unhealthy lifestyle behaviors had higher depression and anxiety scores. There was a statistically significant difference between DASS-21 anxiety and depression score between individuals who had one unhealthy lifestyle behavior and individuals who had two unhealthy lifestyle behaviors. Interestingly, only one individual in the sample had all four unhealthy lifestyle behaviors concurrently, and their DASS-21 scores were the highest in the sample (DASS-21 anxiety: 17, DASS-21 depression: 19). 

## 4. Discussion

The purpose of this study was to explore the relationship between modifiable lifestyle behaviors, including diet quality, sleep duration, alcohol consumption, and tobacco or nicotine use, and symptoms of anxiety and depression in women in college. We did not find significant associations between diet quality, sleep duration, and tobacco or nicotine use and symptoms of mental illness in this sample; however, alcohol consumption was significantly associated with symptoms of mental illness. The overall diet quality of this sample was poor, with only five women reporting an HEI score of ≥80, signifying a high-quality diet. Furthermore, we found that a higher number of unhealthy lifestyle behaviors was associated with greater depression and anxiety symptom severity. Our findings suggest that women in college may benefit from an intervention that targets modifiable lifestyle behaviors, including diet, sleep, alcohol consumption, and tobacco and nicotine, used as part of a comprehensive approach to mental illness treatment and prevention.

College is a time of significant lifestyle change and, thus, represents a time period where encouragement of a healthy lifestyle can help to decrease future disease risk. Individuals in college are more likely to partake in risky or unhealthy behaviors, including poor dietary habits, inadequate sleep, alcohol misuse, and tobacco/nicotine use [11,18,20]. Contrary to these data and the expected behaviors of college students, the majority of this sample met the requirements for sleep, had limited alcohol intake, and were not tobacco or nicotine users. While 45% of the sample did experience symptoms of mental illness (either anxiety and/or depression), the majority of these symptoms fell within the moderate severity range, with only 9% of participants experiencing severe or extremely severe symptoms, which is likely lower than the national average. The 2023 results from the American College Health Association’s national survey found that closer to 24% of women in college report serious psychological distress [52].

### 4.1. Independent Lifestyle Behaviors and Mental Illness

The first aim of this paper was to determine if individual lifestyle behaviors were independently associated with symptoms of mental illness in college women. With regard to diet and mental illness, we did not find a significant difference between HEI score and presence of symptoms of mental illness, which is not consistent with findings from a recent systematic review and meta-analysis [27]. When exploring this connection in prospective cohort studies, Molendijk et al. found that, overall, a higher-quality diet was associated with a reduced risk of symptoms of depression, but not all results supported this claim. Our results may be inconsistent because our sample had poor diets overall, as reflected in the low total HEI score (62.6 ± 10.4). However, previous research exploring the diet quality of women in college in the United States found an even lower HEI score, 57.6 ± 14.5 [53]. In our sample, only 2.3% of participants had good diet quality, and none of the sample fully met the requirements for the Dietary Guidelines for Americans. Future research may benefit from exploring this relationship in samples with more variation in diet quality, specifically more individuals with high diet quality.

Many studies have identified individual dietary components as being associated with symptoms of mental illness [54,55,56,57]. Ultra-processed foods high in saturated fats and sweets have been positively associated with symptoms of depression, whereas foods containing polyunsaturated fatty acids and higher intake of fruits and vegetables have been negatively associated with symptoms of depression [54,55,56,57]. The present study did not find any significant differences between individual HEI components according to the presence of symptoms of mental illness. However, we did find that less than 2% of the sample met the recommendations for whole grains based on the Dietary Guidelines for Americans. In conjunction with the fact that many college students have poor diets, this may reflect the current trends in low-carbohydrate diets as a method for weight loss or weight maintenance. Additionally, our sample reported consuming a high-fat diet, with an average of 36% of calories coming from fat. Similarly, less than 4% of this sample met the recommendations for sodium intake, which may reflect an increased intake of convenience and processed foods, which is common among young adults [11].

Regarding the relationship between sleep and symptoms of mental illness, we did not find a significant association. One third of our sample (30.7%) was not meeting the recommendations from the National Sleep Foundation for young adults, which aligns with the current literature that suggests that sleep suffers in college students [19,48]. Sleep disturbances are often associated with depression and other mental illnesses, like bipolar disorder, due to the role sleep duration and quality plays in the diagnosis of these conditions [33,58,59,60]. The association between sleep disturbances and mental illnesses, such as anxiety and depression, has been thoroughly researched, with multiple studies, including a systematic review, that support a significant association between sleep quality and depression in youth and college students [18,31,32,61]. The present study did not support this hypothesis, which is in alignment with a study from Nyer et al., who did not find an association between sleep disturbances and severity of depression [62]. 

Regarding the relationship between alcohol consumption and symptoms of mental illness, we did find a significant association. Similarly, in a recent meta-analysis and systematic review, Li et al. found that frequent alcohol consumption was associated with an increased risk of depression, while alcohol consumption in general was associated with an increased risk of anxiety [63]. Interestingly, we did not have many heavy drinkers in our sample, with less than 5% of women reporting two or more drinks per day. A previous study in college women found much higher rates of alcohol consumption, with more than 65% of the sample reporting five or more drinks on an occasion [22]. 

With regard to tobacco and nicotine use and symptoms of mental illness, we did not find a significant association. Previous research in college students has found that individuals with mental illness were more likely to be current users of e-cigarettes [29,30]. It is likely that we were not statistically powered to detect associations given that only 9.5% of the sample reported current use of tobacco or nicotine products.

### 4.2. Combined Lifestyle Behaviors and Mental Illness

The second aim of this study was to examine the combined effect of lifestyle behaviors on the presence of symptoms of mental illness in college women. Multiple studies have explored the impact of concurrent lifestyle behaviors on mental illness, but none have looked at this relationship in college students [35,36]. Wang et al. conducted a meta-analysis and found that a healthy lifestyle, or a combination of multiple healthy lifestyle behaviors, was associated with a reduced risk of mental illness [36]. Our study found similar results, with greater symptom severity of anxiety and depression associated with more unhealthy lifestyle behaviors. The Bonferroni post hoc analyses showed that individuals with two unhealthy lifestyle behaviors had statistically significantly higher depression and anxiety scores compared to individuals with one unhealthy lifestyle behavior. Pairwise comparisons did not show any other significant differences, likely due to lack of power in smaller groups, but we did see the highest depression and anxiety scores in the one individual who had all four concurrent unhealthy lifestyle behaviors. Future research may benefit from exploring this dose–response relationship in a larger sample with more diverse behaviors.

### 4.3. Limitations and Future Directions

Our main limitation is the cross-sectional study design; as such, we cannot determine causality or directionality. Many lifestyle behaviors have a bidirectional relationship with mental illness symptoms. Diversity was also lacking in this sample, which limits generalizability to all college students. Although data collection occurred after the COVID-19 pandemic, this study did not address how this period could have influenced the lifestyle behaviors and symptoms of mental illness among participants. Additionally, much of the literature surrounding lifestyle behaviors and mental illness includes physical activity; however, in this study, the relationship between physical activity and symptoms of mental illness was not included. Furthermore, this study did not include other aspects of lifestyle that could contribute to mental illness, such as social support, social engagement, and participation in extracurricular activities.

Future research is necessary to better understand the relationship between lifestyle behaviors and symptoms of mental illness in women in college. Subsequent research should explore this relationship in more diverse samples and include additional lifestyle behaviors. Studies further investigating the effect of concurrent lifestyle behaviors on symptoms of mental illness could help inform future interventions and public policy.

### 4.4. Conclusions

As rates of mental illness among young adults in the United States continue to increase, exploring areas for prevention and intervention is critical. Mental illness among women in college remains a serious public health concern regardless of current treatment methods, including medication and therapy. Studies on the long term effects of psychotropic medications have presented alarming findings that include multiple high-risk side effects, such as an increased risk of suicide (especially in youth) and a lack of long term efficacy in treating symptoms [64,65,66,67,68]. Emerging research into complementary and alternative treatment methods suggests that targeting lifestyle behaviors is safer and more effective than medication at reducing symptom burden in the long term, especially for college students [68,69,70,71]. 

In conclusion, the first aim of this study was to determine if individual lifestyle behaviors were independently associated with symptoms of mental illness in college women. We found a significant association between alcohol consumption and symptoms of mental illness; however, we could not confirm that diet, sleep, and tobacco or nicotine use were independently associated with symptoms of mental illness in college women. The second aim of this study was to examine the combined effect of lifestyle behaviors on symptoms of mental illness in college women. We found a dose–response relationship between concurrent unhealthy lifestyle behaviors and depression and anxiety symptoms. Furthermore, we were able to describe the overall poor-quality diets of college women and identify specific dietary components that need improvement. The present study suggests that women in college may benefit from an intervention that comprehensively targets modifiable lifestyle behaviors, which may ultimately improve symptoms of mental illness.

## Figures and Tables

**Table 1 nutrients-16-02211-t001:** Sociodemographic characteristics according to presence symptoms of mental illness (anxiety and/or depression symptoms) in women in college.

		Presence of Symptoms of Mental Illness		
Characteristics	Total Group (*n* = 222)	With Symptoms of Mental Illness (*n* = 101)	Without Symptoms of Mental Illness (*n* = 121)	*p*-Value
	Mean ± Standard Deviation			
Age (years)	22.8 ± 4.7	22.7 ± 4.5	23.0 ± 4.8	0.637
Body Mass Index (BMI) (kg/m^2^)	24.3 ± 5.9	24.8 ± 6.1	23.9 ± 5.7	0.247
	*n* (%)			
Enrollment Status ^1^				0.084
Full-time	195 (87.8)	85 (84.2)	110 (90.9)	
Part-time	26 (11.7)	16 (15.8)	10 (8.3)	
Program Type ^1^				0.265
Undergraduate	165 (74.3)	79 (78.2)	86 (40.6)	
Graduate	56 (25.2)	22 (21.8)	34 (16.0)	
Employment				0.116
Employed	142 (64.0)	59 (58.4)	83 (68.6)	
Unemployed	80 (36.0)	42 (41.6)	38 (31.4)	
Marital Status				0.672
Single	202 (91.0)	91 (90.1)	111 (91.7)	
Married	20 (9.0)	10 (9.9)	10 (8.3)	
Race				0.346
White	156 (70.3)	68 (67.3)	88 (72.7)	
Asian	31 (14.0)	13 (12.9)	18 (14.9)	
Multi-racial	17 (7.7)	11 (10.9)	6 (5.0)	
Other	8 (3.6)	3 (3.0)	5 (4.1)	
Black	7 (3.2)	4 (4.0)	3 (2.5)	
Native Hawaiian or Pacific Islander	2 (0.9)	0 (0)	2 (1.7)	
American Indian/Alaskan Native	1 (0.5)	0 (0)	1 (0.8)	
Ethnicity ^1^				0.026 *
Not Hispanic or Latino	186 (83.8)	79 (78.2)	107 (88.4)	
Hispanic or Latino	35 (15.8)	22 (21.8) ⇞	13 (10.7) ⇟	
Psychotropic Medication Use ^2^				0.012 *
Yes	61 (27.5)	36 (35.6) ⇞	25 (20.7) ⇟	
No	161 (72.5)	65 (64.4)	96 (79.3)	
Housing Status				0.095
With Family	81 (36.5)	46 (45.5)	35 (28.9)	
Alone or with Roommates	69 (31.1)	27 (26.7)	42 (34.7)	
College Housing	61 (27.5)	22 (21.8)	39 (32.2)	
Other	7 (3.2)	4 (4.0)	3 (2.5)	
Temporarily with Friend or Relative	4 (1.8)	2 (2.0)	2 (1.7)	

* *p*-value of <0.05 determines statistical significance. ^1^ One participant did not answer this question. ^2^ includes antidepressant and anti-anxiety medications. ⇞ Higher proportion than expected. ⇟ Lower proportion than expected.

**Table 2 nutrients-16-02211-t002:** Categorization of depression, anxiety, and stress symptoms based on DASS-21 score in women in college.

DASS-21 Category	Total Group (*n* = 222)
	*n* (%)
DASS-21 Depression	
Normal	141 (63.5)
Mild	35 (15.8)
Moderate	40 (18.0)
Severe	6 (2.7)
Extremely Severe	0 (0)
DASS-21 Anxiety	
Normal	142 (63.9)
Mild	15 (6.8)
Moderate	48 (21.6)
Severe	15 (6.8)
Extremely Severe	2 (0.9)
DASS-21 Stress	
Normal	197 (88.7)
Mild	16 (7.2)
Moderate	9 (4.1)
Severe	0 (0)
Extremely Severe	0 (0)

**Table 3 nutrients-16-02211-t003:** Diet and lifestyle behaviors of participants according to presence of symptoms of mental illness (anxiety and/or depression symptoms) in women in college.

		Presence of Symptoms of Mental Illness		
Characteristics	Total Group (*n* = 222)	With Symptoms of Mental Illness (*n* = 101)	Without Symptoms of Mental Illness (*n* = 121)	*p*-Value
	*n* (%)			
Healthy Eating Index				0.688
Good Diet Quality (HEI ≥ 80)	5 (2.3)	2 (2.0)	3 (2.5)	
Moderate Diet Quality (HEI 51–80)	188 (84.7)	87 (86.1)	101 (83.5)	
Poor Diet Quality (HEI ≤ 50)	29 (13.0)	12 (11.9)	17 (14.0)	
Sleep Duration ^1^				0.181
<7 h	61 (27.5)	34 (33.7)	27 (22.3)	
7–9 h	153 (68.9)	64 (63.4)	89 (73.6)	
>9 h	7 (3.2)	3 (3.0)	4 (3.3)	
Current Tobacco/Nicotine Use				0.084
Yes	21 (9.5)	14 (13.9)	7 (5.8)	
No	200 (90.1)	87 (86.1)	113 (93.4)	
Unknown	1 (0.5)	0 (0)	1 (0.8)	
Alcohol Use				0.049 *
≤1 drink/day	197 (88.7)	85 (84.2)	112 (92.6)	
>1 drink/day	25 (11.3)	16 (15.8) ⇞	9 (7.4) ⇟	
	Mean ± Standard Deviation			
Energy and Nutrients				
Total Energy (kcal)	1896 ± 898	1988 ± 970	1819 ± 830	0.171
Energy (kcal/kg body weight)	30 ± 15	31 ± 17	29 ± 14	0.452
Carbohydrate (%kcal)	49 ± 9	49 ± 9	49 ± 9	0.806
Protein (%kcal)	16 ± 4	15 ± 3	16 ± 4	0.122
Fat (%kcal)	36 ± 3	35 ± 7	36 ± 7	0.366

* *p*-value of <0.05 determines statistical significance. ^1^ One participant did not answer this question. ⇞ Higher proportion than expected. ⇟ Lower proportion than expected.

**Table 4 nutrients-16-02211-t004:** Healthy Eating Index results of participants according to presence of symptoms of mental illness (anxiety and/or depression symptoms) in women in college.

	Standard for Maximum Score (Meets Recommendations of Dietary Guidelines for Americans)	Standard for Minimum Score (Does Not Meet Any Recommendations of Dietary Guidelines for Americans)		Presence of Symptoms of Mental Illness			
Variable			Total Group (*n* = 222)	With Symptoms of Mental Illness (*n* = 121)	Without Symptoms of Mental Illness (*n* = 121)	Meeting Recommendations	*p*-Value
			Mean ± Standard Deviation			*n* (%)	
Healthy Eating Index-2020							
Total Diet Quality (0–100)			62.6 ± 10.4	62.2 ± 9.5	62.9 ± 11.2	0 (0)	0.660
Adequacy Components							
Total Fruits (0–5)	≥0.8 cup equiv./1000 kcal	No Fruit	3.6 ± 1.5	3.6 ± 1.5	3.6 ± 1.5	84 (37.8)	0.846
Whole Fruits (0–5)	≥0.4 cup equiv./1000 kcal	No Whole Fruit	4.5 ± 1.1	4.5 ± 1.0	4.4 ± 1.1	156 (70.3)	0.643
Total Vegetables (0–5)	≥1.1 cup equiv./1000 kcal	No Vegetables	4.2 ± 1.1	4.2 ± 1.0	4.2 ± 1.2	115 (51.8)	0.959
Greens and Beans * (0–5)	≥0.2 cup equiv./1000 kcal	No Dark Green Veg/Legumes	3.8 ± 1.7	3.7 ± 1.7	4.0 ± 1.7	131 (59.0)	0.206
Whole Grains (0–10)	≥1.5 oz equiv./1000 kcal	No Whole Grains	2.4 ± 1.9	2.4 ± 2.0	2.5 ± 1.8	3 (1.4)	0.724
Dairy (0–10)	≥1.3 cup equiv./1000 kcal	No Dairy	5.5 ± 2.8	5.6 ± 2.9	5.4 ± 2.7	26 (11.7)	0.542
Total Protein Foods (0–5)	≥2.5 oz equiv./1000 kcal	No Protein Foods	4.4 ± 1.0	4.3 ± 1.0	4.5 ± 1.0	130 (58.6)	0.336
Seafood/Plant Proteins (0–5)	≥0.8 oz equiv./1000 kcal	No Seafood/Plant Protein	4.0 ± 1.4	4.1 ± 1.4	4.0 ± 1.4	131 (59.0)	0.788
Fatty Acids (0–10)	(PUFA + MUFA)/SFA ≥ 2.5	(PUFA + MUFA)/SFA ≤ 1.2	5.3 ± 3.1	5.2 ± 3.0	5.5 ± 3.3	35 (15.8)	0.581
Moderation Components							
Refined Grains (0–10)	≤1.8 oz equiv./1000 kcal	≥4.3 oz equiv./1000 kcal	7.2 ± 2.7	7.1 ± 2.7	7.2 ± 2.6	45 (20.3)	0.660
Sodium (0–10)	≤1.1 g/1000 kcal	≥2.0 g/1000 kcal	4.1 ± 2.7	4.3 ± 2.8	3.9 ± 2.5	8 (3.6)	0.204
Added Sugars (0–10)	< 6.5% of energy	≥26% of energy	7.7 ± 2.6	7.4 ± 2.5	7.9 ± 2.5	54 (24.3)	0.138
Saturated Fats (0–10)	≤8% of energy	≥16% of energy	5.9 ± 2.9	5.9 ± 2.9	5.8 ± 2.9	27 (12.2)	0.986

* Dark green vegetables or legumes.

**Table 5 nutrients-16-02211-t005:** DASS-21 depression and anxiety scores according to number of concurrent unhealthy lifestyle behaviors.

Variable ^a^	Total Group (*n* = 222)	No Unhealthy Lifestyle Behaviors ^b^ (*n* = 13)	One Unhealthy Lifestyle Behavior (*n* = 122)	Two Unhealthy Lifestyle Behaviors (*n* = 72)	Three or More Unhealthy Lifestyle Behaviors (*n* = 15)	*p*-Value ^c^
Mean ± Standard Deviation						
DASS-21 Depression	7.9 ± 5.8	6.5 ± 6.6	6.9 ± 5.6 ^d^	9.5 ± 5.7 ^e^	9.8 ± 6.1	0.009
DASS-21 Anxiety	6.3 ± 5.1	4.9 ± 4.5	5.0 ± 4.5 ^d^	8.6 ± 5.2 ^e^	7.9 ± 5.2	<0.001

^a^ Range of depression symptoms scores 0–28, range of anxiety symptom scores 0–20. ^b^ Unhealthy lifestyle behaviors include HEI < 80, sleeping <7 or >9 h/day, consuming > 1 alcoholic drink/day, and current use of tobacco or nicotine products. ^c^ One-way analysis of variance. ^d,e^ Values with differing subscripts are statistically significantly different (post hoc, Bonferroni correction *p* < 0.001).

## Data Availability

The datasets presented in this article are not readily available because the research participants were told that their data would not be shared with other researchers or placed in a data repository.

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
