# Peer review of "The Relationship between Lifestyle Behaviors and Mental Illness in Women in College"

_nutrients, 2024, doi:10.3390/nu16142211_

Round 1

Reviewer 1 Report

Comments and Suggestions for Authors

Dear authors,

 I would like to thank you for the opportunity to review this study, which examined the relationship between lifestyle behaviors and mental illness among college women.

Here are my comments:

In the INTRODUCTION chapter, the hypothesis and purpose are correctly stated.

In the MATERIALS AND METHODS chapter, the study design, data collection, and data analysis were done correctly.

In row 176 the reporting of sleep hours is not done correctly. I think you should write that sleep duration < 7 hours and > 9 hours is considered abnormal.

 Chapter RESULTS:

 1 In Table 1, "Enrollment Status" and "Program Type" and "Ethnicity" only 221 subjects are analyzed

2 In Table 3, "Sleep duration" only 221 subjects are analyzed.

When the contribution of carbohydrate (%kcal), protein (%kcal) and fat (%kcal) are analyzed, their contribution provides 101%.

 The DISCUSSION chapter is well documented, but this study has limitations and question marks.

It does not analyze the situation of university women with respect to the COVID-19 period. This period greatly changed people's lifestyles and this certainly influenced the anxiety state of female students.

Not all the factors contributing to people's mental health are analyzed: sports, social relationships, etc.

The CONCLUSIONS chapter should highlight more clearly whether the aim of the study was achieved.

 The references should be updated. Of the 63 references, 30% are written before 2016.

 I wish you good luck!

Author Response

  1. In the INTRODUCTION chapter, the hypothesis and purpose are correctly stated. In the MATERIALS AND METHODS chapter, the study design, data collection, and data analysis were done correctly.
    1. Thank you.
  2. In row 176 the reporting of sleep hours is not done correctly. I think you should write that sleep duration < 7 hours and > 9 hours is considered abnormal.
    1. Thank you for noting this error. We have made this correction. Thank you for your attention to detail.
  3. Chapter RESULTS: 1 In Table 1, "Enrollment Status" and "Program Type" and "Ethnicity" only 221 subjects are analyzed.
    1. Thank you for your attention to detail. We have added a footnote to Table 1 to note that one participant did not answer the questions.
  4. RESULTS: 2 In Table 3, "Sleep duration" only 221 subjects are analyzed.
    1. Thank you for your attention to detail. We have added a footnote to Table 3 to note that one participant did not answer the question.
  5. When the contribution of carbohydrate (%kcal), protein (%kcal) and fat (%kcal) are analyzed, their contribution provides 101%.
    1. We represented these data as whole numbers so due to rounding the percentages add up to 101% (Total group), 99% (With symptoms of mental illness), and 101% (Without symptoms of mental illness).
  6. DISCUSSION chapter is well documented, but this study has limitations and question marks.
    1. Thank you for this comment, we have updated the discussion as noted below.
  7. It does not analyze the situation of university women with respect to the COVID-19 period. This period greatly changed people's lifestyles and this certainly influenced the anxiety state of female students.
    1. We have added the following sentence to the discussion: “Although data collection occurred after the COVID-19 pandemic, this study did not address how this period could have influenced the lifestyle behaviors and symptoms of mental illness among participants.”
  8. Not all the factors contributing to people's mental health are analyzed: sports, social relationships, etc.
    1. We have added the following sentence to the discussion: Furthermore, this study did not include other aspects of lifestyle that could contribute to mental health, such as social support, social engagement, and participation in extracurricular activities.”
  9. The CONCLUSIONS chapter should highlight more clearly whether the aim of the study was achieved.
    1. We have added the following sentences to the conclusion, “In conclusion, the first aim of this study was to determine if individual lifestyle behaviors were independently associated with symptoms of mental illness in college women.” and “The second aim of this study was to examine the combined effect of lifestyle behaviors on symptoms of mental illness in college women.”
  10. The references should be updated. Of the 63 references, 30% are written before 2016.
    1. Thank you for the suggestion, we have incorporated multiple updated references that are highlighted in text and in the reference list.

Reviewer 2 Report

Comments and Suggestions for Authors

The purpose of this study was to explore the relationship between lifestyle factors (diet quality, sleep duration, alcohol consumption, tobacco/nicotine use) and mental illness (anxiety and depression) in women in college. This purpose was accomplished with the following aims: 1. determine if individual lifestyle behaviors were independently associated with  symptoms of mental illness in college women, and 2. examine the combined effect of lifestyle behaviors on symptoms of mental illness in college women.

The manuscript is very interesting, well structured and deals with a topic of great relevance and potential interest for the scientific community. I only have a few minor suggestions for authors.

I believe that in the introductory paragraph, in order to improve the relevance of the manuscript, the authors could better describe the context. In this regard I suggest consulting this recent publication:

Moscatelli et al., Assessment of Lifestyle, Eating Habits and the Effect of Nutritional Education among Undergraduate Students in Southern Italy, Nutrients2023, 15(13), 2894.

In the introductory paragraph, after the purpose of the study, the authors could report their research hypothesis.

Did the authors also record the anthropometric parameters of the recruited subjects? It would be appropriate to include a summary table of the anthropometric parameters of the subjects involved in the study in the methods section.

Author Response

  1. I believe that in the introductory paragraph, in order to improve the relevance of the manuscript, the authors could better describe the context. In this regard I suggest consulting this recent publication: Moscatelli et al., Assessment of Lifestyle, Eating Habits and the Effect of Nutritional Education among Undergraduate Students in Southern Italy, Nutrients, 2023, 15(13), 2894.
    1. Thank you for this suggestion. We have added a sentence into the introduction, “Research suggests that students that live away from their families consume less fruits, vegetables, legumes, and fish than those who live with their families.”
  2. In the introductory paragraph, after the purpose of the study, the authors could report their research hypothesis.
    1. Thank you for this suggestion. We have added a sentence into the introduction, “We hypothesize that lifestyle behaviors, both independently and combined, are associ-ated with symptoms of mental illness in college women.”
  3. Did the authors also record the anthropometric parameters of the recruited subjects? It would be appropriate to include a summary table of the anthropometric parameters of the subjects involved in the study in the methods section.
    1. Data collection for this study occurred online. Participants did self-report height and weight and BMI was calculated (see Table 1). Due to remote data collection, we did not collect any additional anthropometric measures.